# Impact of the Madden–Julian Oscillation on North Indian Ocean Cyclone Intensity

**M. M. Ali** [1,2,*], **Uppalapati Naga Tanusha** [2], **C. Purna Chand** [2], **Borra Himasri** [2], **Mark A. Bourassa** [1] **and Yangxing Zheng** [1]

1 Center for Ocean-Atmospheric Prediction Studies, Florida State University, Tallahassee, FL 32306, USA; bourassa@coaps.fsu.edu (M.A.B.); yzheng@fsu.edu (Y.Z.)
2 Andhra Pradesh State Disaster Management Authority, Kunchanapalli 522501, India; uppalapatitanu@gmail.com (U.N.T.); purnachand.ch@gmail.com (C.P.C.); sri.himaborra@gmail.com (B.H.)
* Correspondence: mmali@coaps.fsu.edu

**Abstract:** The influence of the Madden–Julian Oscillation (MJO) on the intensity of the Tropical Cyclones in the North Indian Ocean is investigated through a machine learning algorithm. The magnitude of wind, considered as a proxy for the intensity, is taken from the Joint Typhoon Warning Centre (JTWC), and the MJO information for 1974–2019 is from Australia's Bureau of Meteorology. These two observations have been collocated and the influence of MJO indices on the wind speed was studied using an artificial neural network technique. The scatter index, defined as the root mean square error (RMSE) normalized to the input data mean, varies from 0.45 for depressions to 0.03 for the super cyclonic storms, indicating that the MJO index is another parameter that should be investigated in cyclone activity studies.

**Keywords:** MJO; cyclone; winds; machine learning





## 1. Introduction

The Madden–Julian Oscillation (MJO), the dominant sub-seasonal variability, prominently represents the intra-seasonal variability in the tropics [1] and bridges weather and climate [2,3]. MJO has a spatial structure of zonal wavenumber with large-scale signals in atmospheric circulation with deep convection propagating eastwards with a speed of ~5 m/s from the Indian Ocean to Pacific Ocean [4]. The MJO is a global tropical mode of variability, with an area of relatively active convection that circles the globe. The MJO influences weather and climate variability, Arctic Oscillations and North Atlantic Oscillations [2], El Niño Southern Oscillations [2,5], the onset of monsoon and monsoon breaks over East Asia [2,4,6], barotropic sea-level variability [7] and Tropical Cyclones (TCs) [6,8–10].

Even though cyclones cannot form during the monsoon season, low pressure areas, depressions and deep depressions are very common in the Indian Ocean in this season. These low pressure areas bring a significant amount of rainfall. Among the several meteorological phenomena, TCs are one of the deadliest and most damaging natural disasters affecting people, livestock, agriculture and the economics of the coastal areas. Hence, this phenomenon and its predictability are worth investigating. Reductions in the uncertainty in TC predictability are of great benefit for the disaster management authorities to plan for the evacuation and mitigation processes [11,12]. This study aims to improve knowledge of TC variability. Thus, it will be of interest from the disaster prevention authorities' point of view. The intensity, a measure of the severity of a cyclone, is primarily estimated from the maximum sustained wind speed and it is one of the major hazards associated with a TC causing damage to houses, bridges, electrical poles, mangroves and the ecosystem. Although the damage in the coastal regions is relatively high, inland damage cannot be ruled out. Wind speed, along with storm track and size, also helps predict storm surge,

which is the most devastating component of the cyclones for coastal India, particularly in regions that have highly varying bathymetry.

While many factors influence the intensity of a cyclone, in this paper, we study the impact of MJO alone without considering the other factors. Cyclone information available from Joint Typhoon Warning Centre (JTWC) during 1974–2019 has been used here to study the impact of MJO on cyclone intensity represented by the one-minute sustained maximum wind magnitude, which is a reasonable diagnostic for wind damage. Girishkumar et al. [13] studied the impact of MJO on the genesis of the North Indian Ocean (NIO) cyclones. Tsuboi and Takemi [8] investigated the interannual relationship between the MJO activity and the TC genesis in the Indian Ocean. The combined impact of ENSO and MJO on the rapid intensification of TCs in the Bay of Bengal was investigated by Girishkumar et al. [13]. Herein, the MJO phase (the location of relatively active weather) and amplitude are defined in terms of dimensionless indices, specifically the real-time multivariate MJO (RMM) indices. These indices (described below) are used in a machine learning algorithm to study the impact of MJO on cyclone wind speeds.

A brief description of the computational procedure by Wheeler and Hendon [14] for MJO quantification is given here. The procedure is mainly based on the computation of EOFs for (i) near-equatorially averaged 850 hPa zonal wind, (ii) 200 hPa zonal wind and (iii) satellite-observed outgoing longwave radiation data. Firstly, the annual cycle and components of interannual variability are removed from the daily observed data. Then, the remaining data are projected onto multiple-variable EOFs, which yields principal component (PC) time series that vary mostly on the intraseasonal time scale of the MJO. This projection acts as an effective filter for the MJO avoiding the restrictions of conventional time filtering while also making the PC time series an effective index for real-time use. The pair of PC time series that form the index are called the Real-time Multivariate MJO series 1 (RMM1) and 2 (RMM2).

Among the machine learning algorithms, artificial neural network (ANN) is one of the popular and powerful data mining tools for computing input-output relationships. It is an information processing paradigm that works somewhat like a hypothesized biological system in the human brain. ANN consists of an interconnected assembly of models, whose functionality is based on a neuron [15]. The analysis can be used as a standalone application or as a complement to statistical analysis. This non-dynamic numerical model has been used in many oceanographic [15–19] and meteorological [19,20] studies. This technique is also found useful for satellite parameter retrievals [16,21–25]. In an ANN model, both the input and the output variables are normalized to vary between 0 and 1.

ANN requires three sets of data: one dataset for training, one for verification and the other for validation. The first dataset is used to train the model, the second dataset to test the model for any shortcomings and finally, the validation dataset, which is held back and not used in developing the model, is used to compare the predicted data with observations and in statistical parameter estimations. Some of the popular formulations of ANN models are Multi-Layer Perceptron (MLP), Radial basis functions (RBF) and Conjugate Gradient Descent models [18,24–27]. Here, we used the MLP method. An MLP generally consists of three nodes (i) an input layer, (ii) a hidden layer and (iii) an output or prediction layer. Each node, except the input layer, is a neuron that uses a nonlinear activation function.

## 2. Data and Methodology

### 2.1. Data

Wind magnitudes (in knots) of the NIO cyclones during 1974–2019 used herein are available from the International Best Track Archive for Climate Stewardship (IBTrACS; Knapp et al., 2010: https://www.ncep.noaa.gov/data/international-best-track-archive-for-climate-stewardship-ibtracs/v04r00/access/csv/, accessed on 29 January 2021). IBTrACS provides information on cyclones from different sources. Here, we used Joint Typhoon Warning Center (JTWC) data alone. These data contain latitude, longitude, surface pressure and wind speed of cyclones. Although these data are available both at 3 and 6 hourly

intervals, we used only 6 hourly interval data in this study. Those cyclone positions which are at irregular intervals were discarded and 239 cyclones were used after this elimination. Daily values of MJO data for the same period were taken from the Bureau of Meteorology, Australia. These data contain latitude, longitude, RMM1 and RMM2, from which its phase (determined from signed values of RMM1 and RMM2) and amplitude (computed as $\sqrt{RMM1^2 + RMM2^2}$) are estimated [14].

*2.2. Methodology*

These MJO values have been collocated with cyclone parameters. Since the cyclone information is generally available four times a day at 0, 6, 12 and 18 h but the MJO values are available only once a day, daily MJO values have been repeated for these four observations of a cyclone. This type of collocation is justified because MJO is of much longer duration compared to the changes in cyclone intensities. These collocated observations have year, month, date, hour, cyclone number, latitude, longitude and wind speed from JTWC and year, month, date RMM1, RMM2, phase and amplitude from the Bureau of Meteorology, Australia. From these collocated observations, 7 datasets have been created for different categories of the cyclonic systems following Mohapatra et al. [28]. The numbers in parentheses represent the number of events. If a cyclone has four observations a day as described earlier, the number of events is 4.

1. All the cyclonic systems which are Depressions (D) and above (231);
2. Deep Depressions (DD) and above (229);
3. Cyclonic Storms (CS) and above (222);
4. Severe Cyclonic Storms (SCS) and above (130);
5. Very Severe Cyclonic Storms (VSCS) and above (74);
6. Extremely Severe Cyclonic Storms (ESCS) and above (34);
7. Super Cyclonic Storms (SUCS) (15).

While 222 CS events and above were formed in the entire study period, SUCS comprised only 15 events. Note that these seven categories are not entirely independent.

ANN analysis has been carried out on these 7 types of datasets. Since ANN needs three sets of data as mentioned earlier, 70% of the data are randomly selected for training, 15% for verification and 15% for validation. We have randomly selected the datasets for training, verification and prediction to avoid any bias that might have crept into the training of the ANN when the training dataset is selected on any other specific criteria. Wind speed is used as a proxy for the intensity of the cyclones. Inputs to the ANN model are latitude, longitude, RMM1, RMM2, amplitude and magnitude of MJO. The outputs are the wind magnitudes of the cyclones. Initially, we trained the model with different combinations of hidden and output neurons and with different algorithms and different cycles. We selected logistic function for the hidden neuron, identity function for the output neuron and Broyden–Fletcher–Goldfarb–Shanno (BFGS) algorithm based on the least root mean square error for the validation dataset. The BFGS algorithm is an iterative method for solving non-linear optimization problems. Finally, a customized neural network is run with the selected functions and algorithms. The BFGS algorithm is run for 20,000 cycles and the ANN model is stopped when the change in error is 0.0000001.

## 3. Results

The statistical results of the ANN model run with wind speed from JTWC as the predictand and month, date, RMM1, RMM2, phase and amplitude of the MJO as the predictors are analyzed. The R-squared, F-statistics and *p*-values between the predictand and the predictors are given in Table 1.

**Table 1.** The statistical results between the predictand (wind speed) and the predictors (month, date, RMM1, RMM2, phase and amplitude).

| Parameter | R-Squared | F-Statistics | *p*-Value |
|---|---|---|---|
| RMM2 | 0.058 | 42.45 | 0.00 |
| Month | 0.041 | 34.21 | 0.00 |
| Amplitude | 0.040 | 34.03 | 0.00 |
| Day | 0.017 | 12.17 | 0.00 |
| RMM1 | 0.014 | 11.84 | 0.00 |
| Phase | 0.010 | 9.95 | 0.00 |

Table 1 gives the statistical relationship between the predictand (wind speed) and the individual predictors (month, date, RMM1, RMM2, phase and amplitude). This analysis is carried out to see how the individual input parameters are related to the wind speed, the output parameter. The information in this table also provides the importance of the input parameters in predicting the wind magnitude of the cyclones. The order of importance of the input variable is identical for the R-squared and F-statistic values. The R-squared values are very small because the cyclone wind magnitudes are controlled by many other meteorological factors other than the MJO indices alone. RMM2 has the highest R-squared value, followed by the month.

Kikuchi et al. [5] concluded that intraseasonal oscillations (ISO) at any time of the year can be expressed in terms of MJO during December-April and boreal summer ISO during June–October. In the Indian Ocean, majority of the cyclones are during April–June and October–January, peaking in November (Figure 1). This figure provides the number of cyclonic systems formed in different months during the study period, providing the temporal distribution of the cyclones. If a system was present in two months, it is considered for both the months. Since most of the systems are present during April–June and from September to December, with a peak in November, the month as an input has the second-highest R-squared value.

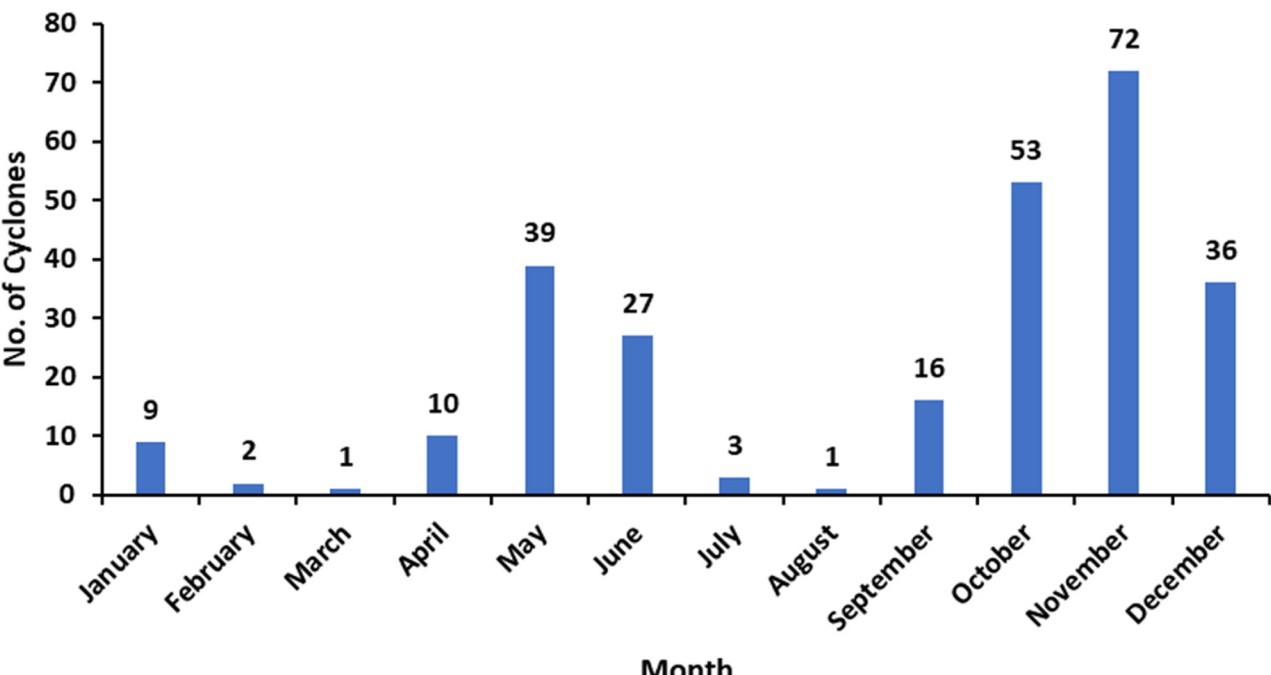

**Figure 1.** Number of cyclonic systems of all categories formed in different months during 1974–2019. The numbers on the bars indicate the number of systems in that month.

The percentage of the deviations calculated from the histograms falling between –80% knots to +80% knots of the validation dataset for all these 7 categories is shown in Table 2. While about 80% of the estimations under the validation category fall between −20% knots to +20% knots for D, DD, CS, SCS and VSCS, 70% of observations fall for ESCS and only 37% for SUCS. Most of the estimations for SUCS fall under 20–40% knots. This could be because the points for training and validation are less for SUCS compared to the other cyclone categories due to which training could not be properly calculated. The statistics for ESCS and SUCS have too large an increase in percentage change to support the suggestion that this result could simply indicate that rapid changes in intensity are relatively rare and that a similar absolute change in intensity results in a smaller percentage change for stronger storms. We later confirm that this simple explanation is incorrect. While this consideration likely contributes to the reduction in percentage change with stronger storms, the explanation for SUCS is likely that such storms can be relatively stable, particularly when the storm becomes annular. Under these conditions, the storms have relatively small absolute changes in intensity. Furthermore, the strongest storms almost always form under near-ideal conditions, and hence are less likely to lose strength interacting with phenomena that inhibit the development of storms (other than encounters with land).

**Table 2.** Percentage of deviations between the estimated and observed values for validation dataset in different categories. The numbers in parentheses represent the total number of observations during all the 4 observation times (at 00, 06, 12, 18 h).

| Percent Change (%) | D (720) | DD (505) | CS (394) | SCS (180) | VSCS (94) | ESCS (39) | SUCS (8) |
|---|---|---|---|---|---|---|---|
| −80 to −60 | 0.14 | 0.00 | 0.00 | 0.00 | 0.00 | 0.00 | 0.00 |
| −60 to −40 | 0.28 | 0.00 | 0.00 | 0.00 | 0.00 | 0.00 | 0.00 |
| −40 to −20 | 8.89 | 9.50 | 8.88 | 6.67 | 8.51 | 17.95 | 0.00 |
| −20 to 0 | 47.22 | 50.89 | 52.28 | 50.56 | 43.62 | 25.64 | 12.50 |
| 0 to 20 | 33.19 | 30.69 | 29.19 | 30.00 | 29.79 | 33.33 | 25.00 |
| 20 to 40 | 7.22 | 6.73 | 8.12 | 11.11 | 14.89 | 17.95 | 62.50 |
| 40 to 60 | 1.81 | 0.99 | 0.51 | 0.56 | 1.06 | 0.00 | 0.00 |
| 60 to 80 | 0.97 | 1.19 | 1.02 | 1.11 | 2.13 | 5.13 | 0.00 |

The cyclones during the study period can be divided into another four categories:

1. Those that have formed over the ocean and dissipated over the land (Figure 2a);
2. Those that have formed over the ocean and dissipated over the oceans with and without crossing the land (Figure 2b);
3. Those that formed over the land moved to the ocean and dissipated (Figure 2c);
4. Those that have formed over land and moved to land after passing over oceans and dissipated over land. The number of cyclones of these four types are 142, 82, 4 and 3, respectively.

Percentage deviations between the observed and MJO-based ANN estimated wind speeds are shown in Figure 3. RMSE and SI (Scatter Index, defined as RMSE normalized to the observed mean value) of the wind magnitudes for the entire dataset and the validation dataset for the 7 categories of cyclones are presented in Table 3. Almost similar values of RMSE and SI for the entire and validation datasets indicate that the random selection of the datasets for the ANN analysis equally covers all the ranges of MJO and wind speeds. For the validation dataset, the RMSE varies from 4.38 knots for the SUCS to 19.25 knots for DD. While RMSE is more useful for emergency managers, the SI is one of the best statistical parameters to judge the skill of the estimations. The SI is highest for D (0.45) and gradually decreases as the cyclone intensity increase and it is the least (0.03) for the SUCS. SI of much less than 1 indicates that the MJO parameters play a significant role in predicting cyclone

intensity. Relatively lower values of RMSE and SI for the entire and validation datasets indicate that the MJO plays a major role for the cyclones with higher intensities.

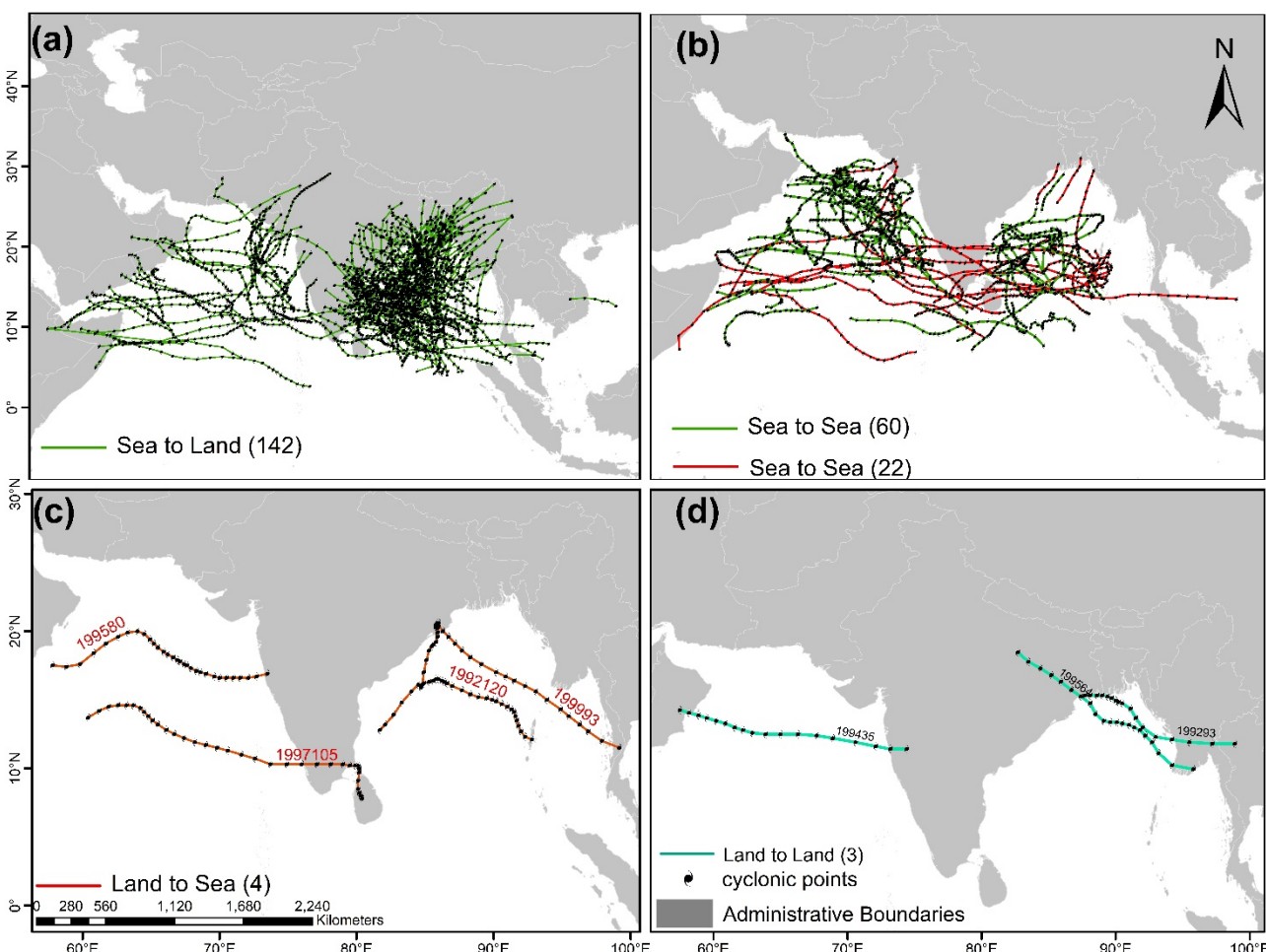

**Figure 2.** Cyclone tracks that have (**a**) formed over the sea and dissipated over the land (no. of cyclones 142; green in color); (**b**) formed over the sea and dissipated over the sea itself without crossing the land (no. of cyclones 60; red in color), formed over the sea and dissipated over the sea after crossing the land (no. of cyclones 22; green in color); (**c**) formed over land and dissipated over sea (no. of cyclones 4; red in color) and (**d**) formed over land and dissipated over land (no. of cyclones 3; green in color).

**Table 3.** Statistical parameters for RMSE and SI for all the categories for the entire and validation datasets alone.

|  | All | | | Validation | | |
|---|---|---|---|---|---|---|
|  | Count | RMSE (Knots) | SI | Count | RMSE (Knots) | SI |
| D | 4805 | 18.26 | 0.45 | 720 | 18.38 | 0.45 |
| DD | 3369 | 17.66 | 0.38 | 505 | 19.25 | 0.41 |
| CS | 2627 | 18.07 | 0.34 | 394 | 19.16 | 0.36 |
| SCS | 1201 | 17.19 | 0.24 | 180 | 17.08 | 0.24 |
| VSCS | 629 | 15.14 | 0.17 | 94 | 13.66 | 0.16 |
| ESCS | 261 | 9.87 | 0.09 | 39 | 10.22 | 0.09 |
| SUCS | 59 | 5.69 | 0.04 | 8 | 4.38 | 0.03 |

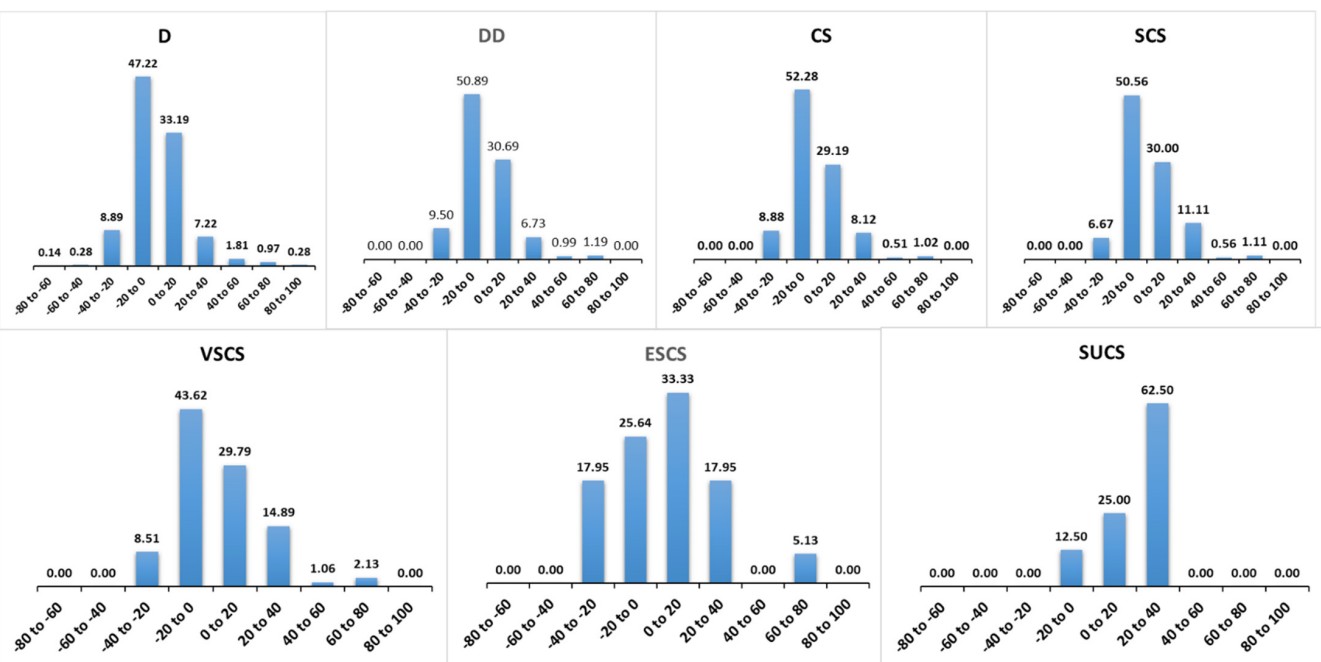

**Figure 3.** Percentage of deviations for validation datasets divided into categories of D, DD, CS, SCD, VSCS, ESCS and SUCS.

The lower SI values for higher intensity categories indicate that the MJO-based ANN forecast skill improves with the increasing intensity of cyclonic system categories. This result implies that MJO parameters have more impact on more intense cyclones.

Lafleur et al. [29] discussed the active role played by MJO amplitudes on cyclone intensities. As discussed by Ref. [29], we divided the dataset into 4 categories based on MJO strength as: (i) all the amplitudes; (ii) amplitudes 1–1.5; (iii) amplitudes 1.5–2.0 and (iv) amplitudes >2.0. The SI decreases with increasing amplitude for all categories of the cyclonic systems showing the important role played by amplitude in cyclone intensity prediction. Wheeler and Hendon [14] observed MJO to be more active when the amplitude is >2.0. However, they did not study its influence on different types of cyclone categories, as is done herein. Besides the SI decreasing with increasing cyclone amplitude, it also decreases with the increasing intensity of the cyclonic systems. This implies that MJO influences more for high intensity cyclones. We studied the influence of the amplitude on cyclone intensity for (i) all phases together and (ii) for phases 1, 2, 3, 4 covering the Indian Ocean (2, 3) and the two adjoining phases (1, 4). In addition, all the phases have been further divided into different amplitudes as (i) all amplitudes, (ii) amplitudes ≥1, (iii) amplitude ≥1.5 and (iv) amplitude ≥2. The influence of MJO phases and amplitudes, thus categorized, on the wind speed of the validation dataset is given in Table 4. We did not compute SI for ESCS, and SUCS as the number of points is too small (when the phases are divided into different categories) to meaningfully compute SI. The SI is least for amplitudes >2 which implies that the impact of MJO on cyclone intensity is more for greater MJO amplitudes.

The number of cyclones formed when the MJO index is in phases 1, 2, 3, 4 is 155 while that formed in phases 5, 6, 7 or 8 is 76. This indicates that when the MJO is in the Indian Ocean, phases of 2 or 3 and the two adjoining phases of 1 or 4, the conditions are more conducive to the formation of cyclones.

**Table 4.** Influence of MJO amplitude on estimated wind speed scatter index (SI) for all the categories of the cyclonic systems for (i) all amplitudes, (ii) amplitudes ≥1, (iii) ≥1.5 and (iv) ≥2 separately for the validation dataset The number of points for each category is also provided.

| **All Phases** | | | | | | | | |
|---|---|---|---|---|---|---|---|---|
| | **Scatter Index (SI)** | | | | **Number of Points** | | | **Correlation Coefficient** |
| **Category/Phase** | **All** | **≥1** | **≥1.5** | **≥2** | **All** | **≥1** | **≥1.5** | **≥2** | **All** |
| D | 0.485 | 0.476 | 0.467 | 0.323 | 720 | 457 | 255 | 110 | 0.431 |
| DD | 0.428 | 0.428 | 0.383 | 0.278 | 505 | 328 | 189 | 84 | 0.423 |
| CS | 0.381 | 0.387 | 0.345 | 0.264 | 394 | 253 | 148 | 66 | 0.427 |
| SCS | 0.271 | 0.263 | 0.208 | 0.138 | 180 | 118 | 76 | 33 | 0.467 |
| VSCS | 0.183 | 0.152 | 0.149 | 0.085 | 94 | 63 | 40 | 21 | 0.638 |
| **Phases—1, 2, 3, 4** | | | | | | | | |
| | **Scatter Index (SI)** | | | | **Number of Points** | | | **Correlation Coefficient** |
| **Category/Phase** | **All** | **≥1** | **≥1.5** | **≥2** | **All** | **≥1** | **≥1.5** | **≥2** | **All** |
| D | 0.473 | 0.463 | 0.447 | 0.430 | 432 | 291 | 170 | 77 | 0.435 |
| DD | 0.408 | 0.396 | 0.352 | 0.252 | 310 | 217 | 130 | 58 | 0.503 |
| CS | 0.370 | 0.342 | 0.310 | 0.248 | 240 | 166 | 102 | 43 | 0.478 |
| SCS | 0.255 | 0.259 | 0.206 | 0.134 | 110 | 74 | 50 | 21 | 0.605 |
| VSCS | 0.174 | 0.174 | 0.121 | 0.055 | 53 | 35 | 24 | 13 | 0.705 |

## 4. Summary and Conclusions

The influence of the MJO on the intensity of the Tropical Cyclones in NIO is investigated using an ANN approach. The MJO is known to influence the weather phenomena and the ANN is one of the popular machine learning algorithms for computing input-output relationships. For this purpose, daily 6 hourly values of the intensity of the cyclones in the NIO are obtained from JTWC and MJO indices from the Bureau of Meteorology, Australia. Wind magnitude is the predictand and the MJO indices (RMM1, RMM2, magnitude, amplitude along with the positional information) are the predictors for the ANN model. Almost similar values of RMSE and SI between the observed and predicted values for the training, selection and validation datasets indicate that the random selection of the datasets for the ANN analysis equally covers all the ranges of MJO and wind speeds. Statistical analysis through a machine learning process indicates that MJO index influences the intensity of the cyclones in the north Indian Ocean. The deviations between the wind speeds estimated using the month, date, RMM1, RMM2, amplitude and phase of the MJO and the actual observations fall between −20 knots and +20 knots for 80% of the cases for D, DD, CS, SCS and VCCS categories of the cyclone. However, 70% and only 37% of ESCS and SUCS respectively fall under this range. Super cyclonic storms can be relatively stable, particularly when the storm becomes annular. Under these conditions, the storms have relatively small absolute changes in intensity. This could be one of the reasons for these storms not being significantly influenced by MJO indices. However, the SI of the validation dataset reduces from 0.45 for D to 0.03 for SUCS, implying that MJO has more influence on the cyclones with more intensity. Both MJO phase and amplitude influence the cyclone intensity. The cyclonic systems formed when the MJO is in phases 1, 2, 3 or 4 are double those formed during the 5, 6, 7 or 8 phases. The influence of MJO amplitude on wind speed increases with increasing amplitude. Here, we statistically studied the influence of MJO alone on cyclone intensities. More detailed studies, particularly using dynamical modeling, are required to come to a more physically-based conclusion.

**Author Contributions:** Conceptualization, M.M.A.; methodology, C.P.C., B.H.; validation, M.M.A.; formal analysis, C.P.C., B.H.; data curation, U.N.T.; writing—original draft preparation, M.M.A.; writing—review and editing, M.A.B., Y.Z. and C.P.C. All authors have read and agreed to the published version of the manuscript.

**Funding:** This research received no external funding.

**Institutional Review Board Statement:** Not applicable.

**Informed Consent Statement:** Not applicable.

**Data Availability Statement:** All the data used in this study are obtained from the two public sources mentioned in the acknowledgement.

**Acknowledgments:** The authors thank their respective organizations for their support and encouragement. Wind magnitudes of the NIO cyclones were obtained from the International Best Track Archive for Climate Stewardship (https://www.ncep.noaa.gov/data/international-best-track-archive-for-climate-stewardship-ibtracs/v04r00/access/csv/, accessed on 29 January 2021). Information on MJO was available at www.bom.gov.au/climate/mjo/graphics/rmm.74toRealtime.txt, accessed on 29 January 2021. The authors thank both the organizations for the data. The authors also thank the anonymous referees for their critical but constructive comments.

**Conflicts of Interest:** The authors have no conflict of interest.

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
