# Peer review of "Impact of the Madden–Julian Oscillation on North Indian Ocean Cyclone Intensity"

_atmosphere, doi:10.3390/atmos12121554_

Round 1
Reviewer 1 Report
The paper has a permanent interest from the disaster prevention authorities to plan the evacuation and mitigation process. The publication of the paper is suitable as the “Atmosphere” Following small revisions are necessary before the publish.
- The Madden-Julian Oscillation looks a large-scale atmospheric circulation in the Indian Ocean. Could you please make a simple image figure of MJO, if possible?
- Line 86; What is the unit of RMM1 and RMM2 ?
- Figure 1; The number of cyclones is coming from the number of those generated in the Indian Ocean?
- Line 137; “80% m/s to 80% m/s for …” is right?
- Line 209 and 210; Table 4 is put between Line 209 and 210.
Author Response
We thank the reviewer for the constructive comments. We gave point-wise replies to comment in track changes in the attached file.

Reviewer 2 Report
Review of the manuscript atmosphere-1444655
The manuscript can be very interesting but is very confusing to read, lacks data and model information, the summary, introduction and discussion/conclusion sections are too short.
Please add a page of text describing the method ANN /MLP, how the method was introduced into the manuscript, how ANN /MLP was set up and run? It is not clear what input data was used, what the starting parameters are, how many neurons and layers the model contains. What software was used for the analysis. It is difficult for me to analyse the results if I do not know how the model is set up.
What are the results of the ANN /MLP method? (Table 1). Please add more text showing the results of the model.
How does Figure 1 relate to the ANN /MLP method?
The subsection about the data is too short, please add more information, maybe a table showing what data was used.
RMM1 & 2 not defined. L86-88, hard to read, please expand.
Author Response
The authors thank the reviewer for his critical but constructive comments. Replies to each comment are given in the attached file.

Round 2
Reviewer 2 Report
I have no more coments
Author Response
The second referee has no comments/suggestions.